# Telehealth at scale can improve chronic disease management in the community during a pandemic: An experience at the time of COVID-19

Stefano Omboni[1,2]*, Tommaso Ballatore[1‡], Franco Rizzi[1‡], Fernanda Tomassini[1‡], Edoardo Panzeri[1☯], Luca Campolo[1☯]

1 Clinical Research Unit, Italian Institute of Telemedicine, Varese, Italy, 2 Department of Cardiology, Sechenov First Moscow State Medical University, Moscow, Russian Federation

☯ These authors contributed equally to this work.
‡ These authors also contributed equally to this work.
* stefano.omboni@iitelemed.org

**Data Availability Statement:** All relevant data are within the manuscript and its Supporting information files.

## Abstract

### Background

During the COVID-19 pandemic, telehealth became a vital resource to contain the virus's spread and ensure continuity of care of patients with a chronic condition, notably arterial hypertension and heart disease. This paper reports the experience based on a telehealth platform used at scale to manage chronic disease patients in the Italian community.

### Methods and findings

Patients' health status was remotely monitored through ambulatory blood pressure monitoring (ABPM), resting or ambulatory electrocardiogram (ECG), spirometry, sleep oximetry, and cardiorespiratory polysomnography performed in community pharmacies or general practitioners' offices. Patients also monitored their blood pressure (BP), heart rate (HR), blood oxygen saturation ($SpO_2$), body temperature, body weight, waist circumference, blood glucose, and lipids at home through a dedicated smartphone app. All data conveyed to the web-based telehealth platform were used to manage critical patients by doctors promptly. Data were analyzed and compared across three consecutive periods of 2 months each: i) before the national lockdown, ii) during the lockdown (from March 9 to May 17, 2020), and iii) after the end of the containment measures. Overall, 13,613 patients visited community pharmacies or doctors' offices. The number of overall tests dropped during and rose after the lockdown. The overall proportion of abnormal tests was larger during the outbreak. A significant increase in the prevalence of abnormal ECGs due to myocardial ischemia, contrasted by a significantly improved BP control, was observed. The number of home users and readings exchanged increased during the pandemic. In 226 patients, a significant increase in the proportion of $SpO_2$ readings and a significant reduction in the entries for all other parameters, except BP, was observed. The proportion of abnormal $SpO_2$ and BP

**Funding:** The authors received no specific funding for this work.

**Competing interests:** I have read the journal's policy and the authors of this manuscript have the following competing interests: SO is scientific consultant of Biotechmed Ltd, provider of telehealth services. The other authors declare no conflicts of interest regarding the publication of this paper.

values was significantly lower during the lockdown. Following the lockdown, the proportion of abnormal body weight or waist circumference values increased.

## Conclusions

Our study results support the usefulness of a telehealth solution to detect deterioration of the health status during the COVID-19 pandemic.

## Trial registration

The study is registered with ClinicalTrials.gov at number NCT03781401.

## Introduction

Italy has been particularly affected by the COVID-19. Since the first death due to severe acute respiratory syndrome on February 21, 2020, the number of infected people rapidly increased, and the Country paid a high toll of excess deaths [1]. Fatalities occurred mainly among older adults and subjects with underlying multiple chronic conditions, notably arterial hypertension and heart disease [2]. To stem the disease's spread, the Italian Government implemented extraordinary containment measures with a general lockdown between March 9 and May 17, 2020. During this period, people were confined at home, while hospitals and family doctors strived to manage infected people. Hence, patients with pre-existing chronic conditions were often left behind with a consequent worsening of their status and excess mortality from causes other than the COVID-19 [3–5].

During the recent coronavirus disease outbreak, telehealth thrived and emerged worldwide as an indispensable resource to improve patients' surveillance, curb the spread of the virus, and favor early identification and prompt management of infected people [6–8]. Telehealth also helped ensure continuity of care of vulnerable patients with multiple chronic conditions, such as arterial hypertension, heart disease, chronic obstructive pulmonary disease (COPD), and diabetes [9–13].

In Italy, in the course of the first wave of the COVID-19 pandemic, available telehealth solutions have been used to track and promptly detect potentially infected individuals [14, 15] and monitor patients with any critical chronic condition (including heart failure, diabetes, oncological disease, skin disease, neurological or psychiatric disease, immunological disease) restrained at home [16–22]. However, preliminary reports suggest that the Country was mostly unprepared to implement existing telehealth solutions for this purpose [11, 23–26]. That occurred because experiences with telehealth in Italy are primarily anecdotal, often limited to local projects or studies, and only in rare instances is telehealth implemented at scale [24].

This paper will present our experience during the pandemic based on telehealth interventions delivered to patients nationwide by community pharmacies or general practitioners or at their dwellings.

## Materials and methods

### Study population and design

The population of this analysis was based on that of the TEMPLAR Project, an observational, cross-sectional, retrospective, multicenter study involving community pharmacies and general practitioners distributed over the whole Italian territory. The study is registered with

ClinicalTrials.gov at number NCT03781401 and with the Registry of Patient Registries (RoPR) at number 41818. Details on the study design and procedures are available in the main study publication [27]. Two different sets of data were considered for this analysis. The first dataset included individuals referred to local pharmacies or general practitioners' offices for specific professional diagnostic tests through a telehealth platform. The second dataset encompassed patients monitoring various parameters at home through a smartphone app connected to the same telehealth platform used by the healthcare professionals (for further technical details, see below). Data were collected between December 31, 2019, and July 26, 2020. The study included subjects of either sex and any age i) presenting to community pharmacies with a general practitioner's prescription and guidelines-based clinical indications for the test, or ii) managed directly by their general practitioners, or iii) downloading the app from the telehealth website and using it connected with their referring doctor. The study was conducted according to the principles of the Declaration of Helsinki. At the time of the test, informed consent was obtained electronically by the pharmacist from each individual to use subjects' pseudo-anonymized data for aggregated analysis, according to the current European General Data Protection Regulation [28]. The pharmacist was previously appointed data processor by the controller (the telemedicine platform manager) with an act drawn up according to the data protection code, and he was thus authorized to collect patient's sensitive data for the purpose of obtaining the ABPM. No approval from any Ethics Committee was required since the tests were performed as part of routine clinical practice (primary purpose of data processing) and not in the context of a research study. When the test was performed, the healthcare professional or the patient inputted any information about concomitant ailments and medications in the telehealth software (website or smartphone app). All data were anonymized before any analysis.

## The web-based telehealth platform

The telehealth platform used in this study allows monitoring patients' health status either in a professional or home setting, providing medical counseling and teleconsultations to final users [29]. Diagnostic tests are obtained through various clinically validated and certified medical devices that enable collecting specific vital and non-vital parameters (for further details, see below). The professional diagnostic tests are uploaded on the web-based telehealth platform through a personal computer. Parameters measured by patients at home with their own medical devices are collected through a mobile app running on an Android™ smartphone or tablet. The app can be freely downloaded from the Google play store. Once collected or inputted, data are transferred to the server host through the Internet and analyzed by the core software using specific, clinically validated, and certified algorithms. In the case of professional tests, an automatic test report is generated and then checked, verified, and signed by a physician. The medical report is then forwarded to the patient through the telehealth web platform, mobile app, or e-mail. In the case of data collected at home, the mobile app provides immediate feedback to the user about the parameter's critical level. It also allows setting reminders for daily drug intake and recording adherence to medications. All these data are also made available on the web portal, where they can be viewed by an authorized physician who can provide a medical opinion.

## Type of tests and devices

Professional tests included: i) 24-hour ambulatory blood pressure monitoring (ABPM) through an automated electronic upper-arm BP monitor (Microlife WatchBP O3), ii) electrocardiogram (ECG) at rest through a 12-lead hand-held electrocardiograph (Beneware CS-280), iii) 24- or 48-hour Holter ECG through a 3-lead electrocardiograph (Beneware CT-08S), iv) spirometry (MIR Spirodoc), iv) sleep oximetry (MIR Spirodoc), v) cardiorespiratory

polysomnography with a 7-signal device (Philips Alice NightOne) [29]. Home data were collected with patients' own devices and were manually inputted in the mobile app. The mobile app could also download the data directly from the device with no need for manual input, if medical devices equipped with Bluetooth technology were used. However, in the case of this analysis, no users owned a wireless device. Automated electronic upper arm BP monitors, pulse oximeters, electronic thermometers, electronic scales, glucometers, and lipid meters were used to detect and record BP, heart rate (HR), blood oxygen saturation ($SpO_2$), body temperature, body weight, blood glucose, and lipids. Waist circumference measured with a tape measure could also be entered in the app.

## Statistical analysis

The primary analysis was performed on data collected between December 31, 2019, and July 26, 2020. This time interval was selected to obtain three 2-month comparable subperiods centered on the pandemic's peak when the Italian Government ordered a general country lockdown: i) a pre-lockdown period (December 31, 2019 –March 8, 2020), ii) a lockdown period (March 9, 2020 –May 17, 2020), and iii) a post-lockdown period (May 18, 2020 –July 26, 2020).

For each period, the number of professional diagnostic tests, the number of home readings, and the proportion of abnormal tests or readings were computed. ABPMs were categorized as abnormal in the case of an elevated 24-hour (systolic blood pressure, SBP $\geq$130 mmHg and/or diastolic blood pressure, DBP $\geq$80 mmHg) or day-time (SBP $\geq$135 mmHg and/or DBP $\geq$85 mmHg) or night-time average BP (SBP $\geq$120 mmHg and/or DBP $\geq$90 mmHg) [30]. ECGs were classified as abnormal when at least one of the following was reported and validated by the doctor: ectopic beats, sustained supraventricular or ventricular arrhythmias, fascicular blocks, atrioventricular blocks, myocardial ischemia, ventricular or atrial hypertrophy. Lung function was classified as abnormal when the spirometry showed an obstructive or restrictive disease or when the sleep oximetry or cardiorespiratory polysomnography identified an obstructive sleep apnea syndrome. Single readings obtained at home were categorized as abnormal according to the following thresholds: i) SBP $\geq$135 mmHg and/or DBP $\geq$85 mmHg, ii) HR >100 bpm, iii) body mass index (BMI) $\geq$25 kg/m$^2$ and/or waist circumference $\geq$94 cm in men and $\geq$80 cm in women, iv) $SpO_2$ <96%, v) body temperature >37.5˚C, vi) total cholesterol >190 mg/dL and/or low density lipoprotein (LDL) cholesterol $\geq$115 mg/dL and/or triglycerides $\geq$150 mg/dL and/or high density lipoprotein (HDL) cholesterol <40 mg/dL in men and/or <50 mg/dL in women and/or blood glucose $\geq$100 mg/dL [31–33]. The adherence to antihypertensive treatment was measured by the voluntary recording of the daily drug intake on the app.

The study population's clinical and demographic characteristics and the proportion of abnormal tests or readings were computed and compared across the three study subperiods. Differences were estimated by analysis of variance (ANOVA) or Chi-square test. A secondary confirmatory analysis was performed by comparing the three subperiods of 2020 with the corresponding periods of 2019 (December 31, 2018, and July 28, 2019, in the absence of coronavirus outbreak). The level of statistical significance was set at 0.05. Data are shown as mean ±SD or 95% confidence interval for continuous variables and absolute (n) and relative (%) frequencies for discrete variables. Data management and analysis were carried out by SPSS for Windows version 20.

## Results

### Community pharmacies and general practitioners' offices

In the period between December 31, 2019, and July 26, 2020, a total of 13,613 patients visited 681 community pharmacies or general practitioners' offices and underwent a single ECG

**Table 1. General characteristics of subjects and tests performed in the community during the COVID-19 pandemic.**

| Pharmacies and general practitioners' offices | Pre-lockdown | Lockdown | Post-lockdown | p-value |
|---|---|---|---|---|
| Operators | 637 | 279 | 528 | 0.0001 |
| Total subjects (n) | 7758 | 874 | 4981 | 0.0001 |
| Age (years, mean±SD) | 48.2 ± 23.1 | 58.2 ± 18.9 | 55.3 ± 22.5 | 0.0001 |
| Sex | | | | |
| Male (n, %) | 3374 (43.5) | 425 (48.6) | 2264 (45.5) | 0.004 |
| Female (n, %) | 4384 (56.5) | 449 (51.4) | 2717 (54.5) | |
| Antihypertensive treatment (n, %) | 1027 (13.2) | 137 (15.7) | 579 (11.6) | 0.001 |
| Cardiovascular disease (n, %) | 1237 (15.9) | 209 (23.9) | 1072 (21.5) | 0.0001 |
| Cardiovascular risk factors (n, %) | 2029 (26.2) | 305 (34.9) | 1487 (29.9) | 0.0001 |
| Concomitant diseases, symptoms or treatments (n, %) | 3498 (45.1) | 571 (65.3) | 2821 (56.6) | 0.0001 |
| Area of the Country | | | | |
| North (n, %) | 2957 (38.1) | 313 (35.8) | 1894 (38.0) | 0.0001 |
| Center (n, %) | 1344 (17.3) | 99 (11.3) | 735 (14.8) | |
| South (n, %) | 3457 (44.6) | 462 (52.9) | 2352 (47.2) | |
| Abnormal test results (n, %) | 3566 (46.0) | 536 (61.3) | 2628 (52.8) | 0.0001 |
| **Home users** | **Pre-lockdown** | **Lockdown** | **Post-lockdown** | **p-value** |
| Total subjects (n) | 21 | 170 | 61 | 0.0001 |
| Transmitted readings (n) | 350 | 2838 | 1602 | 0.0001 |
| Age (years, mean±SD) | 54.5 ± 14.9 | 49.1 ± 14.0 | 52.6 ± 17.4 | 0.132 |
| Sex | | | | |
| Male (n, %) | 15 (71.4) | 116 (68.2) | 46 (75.4) | 0.571 |
| Female (n, %) | 6 (28.6) | 54 (31.8) | 15 (24.6) | |
| Concomitant diseases or treatments (n, %) | 5 (23.8) | 31 (18.2) | 10 (16.4) | 0.750 |
| Abnormal values (n, %) | 94 (26.9) | 510 (18.0) | 265 (16.5) | 0.028 |

Data are shown according to the setting (pharmacies or general practitioners' offices vs. home) and period (before, during, and after the lockdown).

(n = 9,466), ABPM (n = 4,110), or lung function test (n = 37). As summarized in Table 1 and Fig 1, the number of tests dramatically dropped during the lockdown compared to the pre-lockdown period.

During the lockdown period, patients performing tests were older and more often males than during the period preceding the lockdown. They were also more often under pharmacological treatment, including antihypertensive drugs, or affected by a concomitant disease, including cardiovascular disease. The tests were more frequently performed in the Country's southern regions, where the virus spread was limited.

The proportion of abnormal tests was significantly larger through the lockdown than in the previous weeks (Table 1 and Fig 2).

Following the lockdown, the tests increased in number, albeit to values lower than those preceding the restriction. The abnormal tests' frequency was still high and significantly more prominent after than prior to the lockdown.

As depicted in Fig 3, during the lockdown, there was a significant increase in the frequency of ambulatory ECGs and a reduction in the frequency of resting ECGs.

Following the lockdown, the frequency of ambulatory ECGs remained higher than before the outbreak, while resting ECGs rose to percentages close to pre-lockdown values. The frequency of ABPMs was stable before and during the lockdown but significantly dropped following the restriction. During the lockdown, 24-hour BP control improved mainly due to enhanced day-time BP control, whereas night-time BP control worsened. A significant

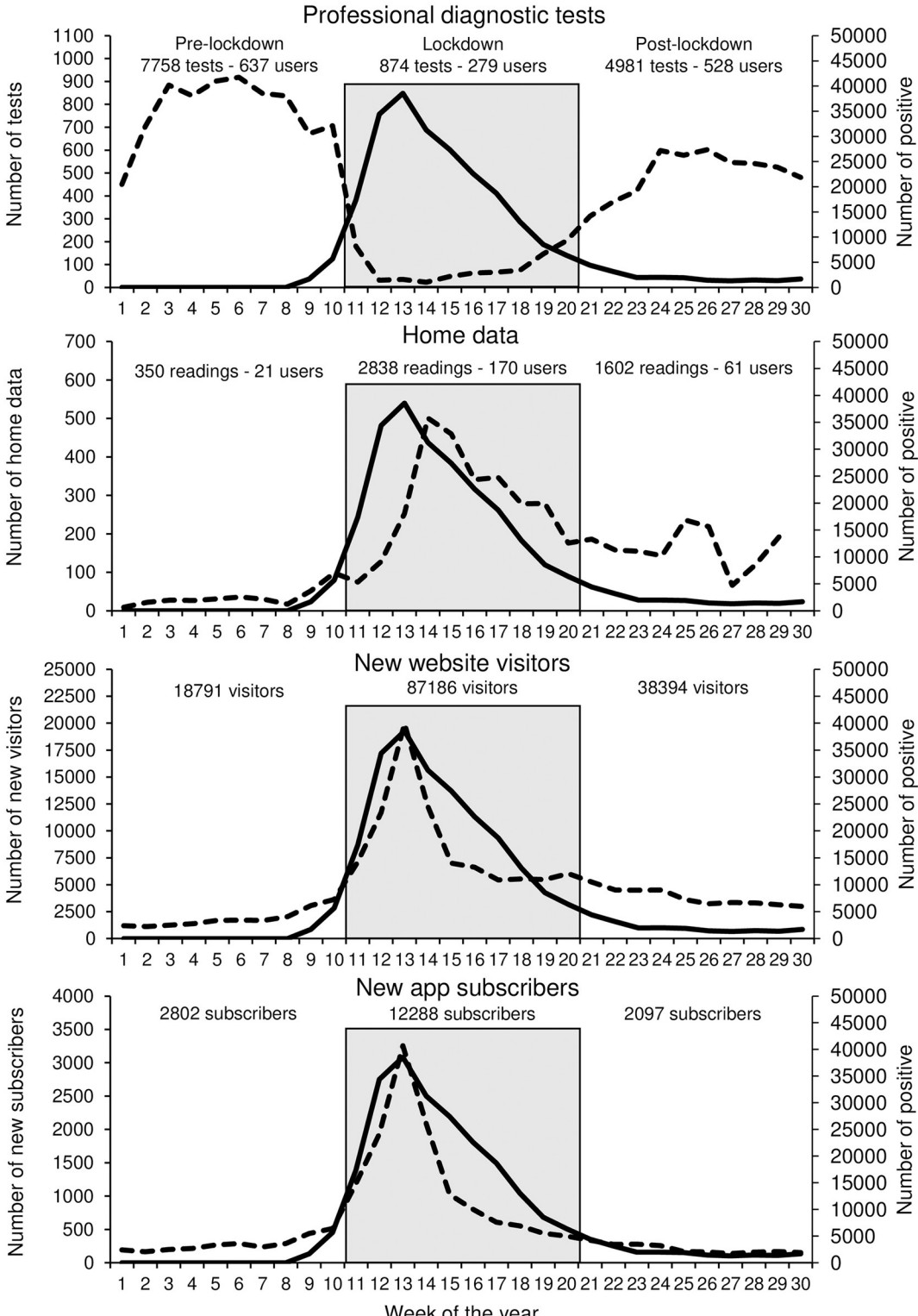

**Fig 1. Weekly number of professional tests, home data, new visitors to the telemedicine website, and new app subscribers (dashed lines).** The thick continuous line represents the number of positive to SARS-CoV-2 tests. Pre-lockdown, lockdown, and post-lockdown periods are also identified.

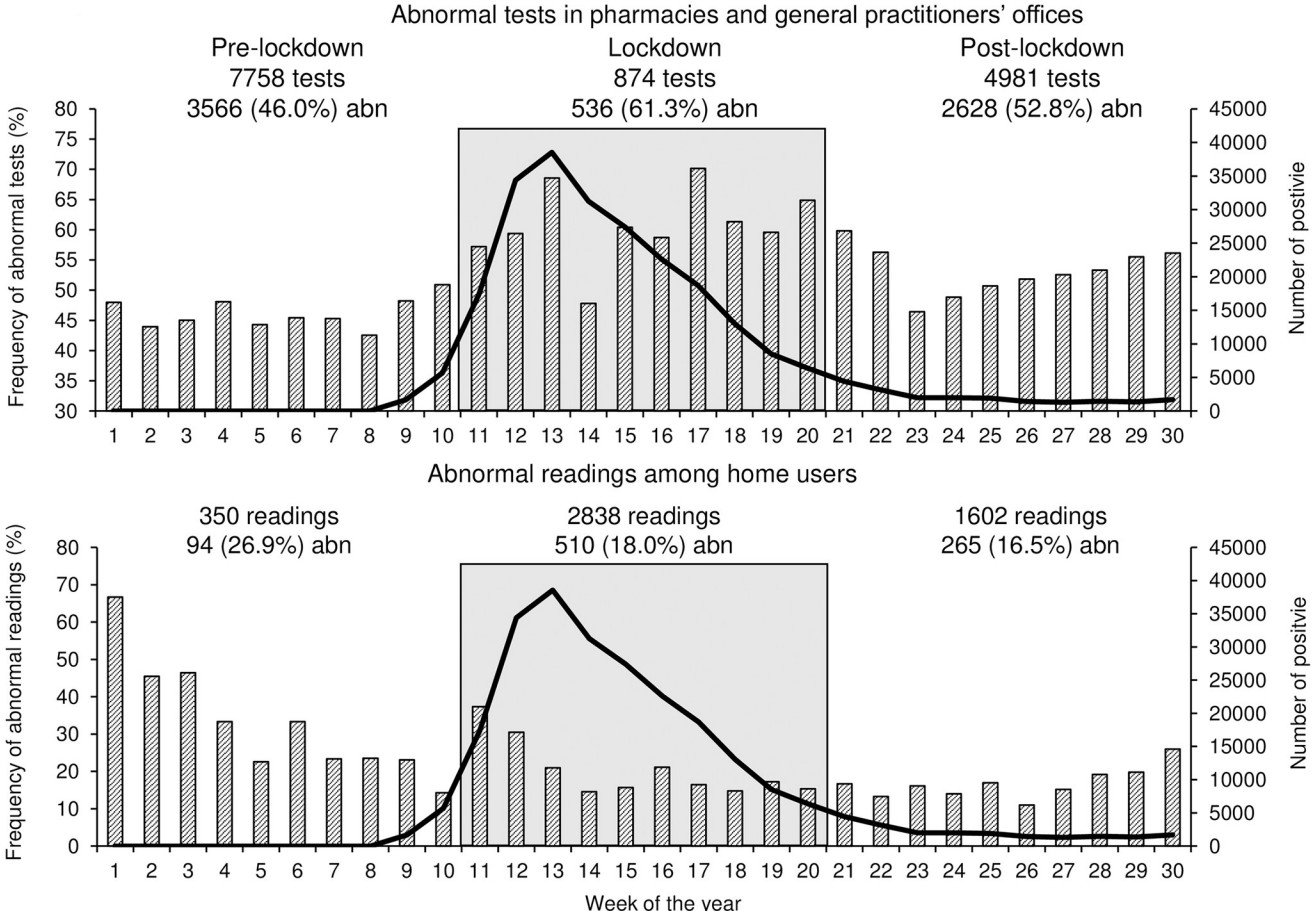

**Fig 2. Weekly frequency (%) of abnormal tests in pharmacies and general practitioners' offices or abnormal readings collected at home (striped bars).** The thick continuous line represents the number of positive to SARS-CoV-2 tests. Pre-lockdown, lockdown, and post-lockdown periods are also identified.

increase in the proportion of abnormal ECGs was observed during and to a lower extent after the lockdown. The principal ECG diagnosis was myocardial ischemia. All patients with clinically significant abnormal tests of any kind were referred to the necessary care.

As far as lung tests are concerned, their size was limited compared to the other tests. However, a reduced frequency of abnormal tests was observed during the lockdown.

## Home users

A total of 226 single subjects used the app during the observation period. The number of new visitors to the telehealth platform's website and the number of new subscribers to the app, as well as the number of home users and readings exchanged, markedly and significantly increased during the lockdown (Table 1 and Fig 1). Conversely to what was observed for patients visiting community pharmacies and general practitioners' offices, users were younger during the lockdown. No differences were observed across the three periods regarding sex, medication use, and disease distribution. The proportion of patients treated with antihypertensive medications did not significantly vary during and after the lockdown (2, 6.3% vs. 14, 9.5% before the lockdown; p = 0.635). However, following the lockdown, an increase in the frequency of initiation or adjustment of antihypertensive medications (35.7% of subjects vs. none

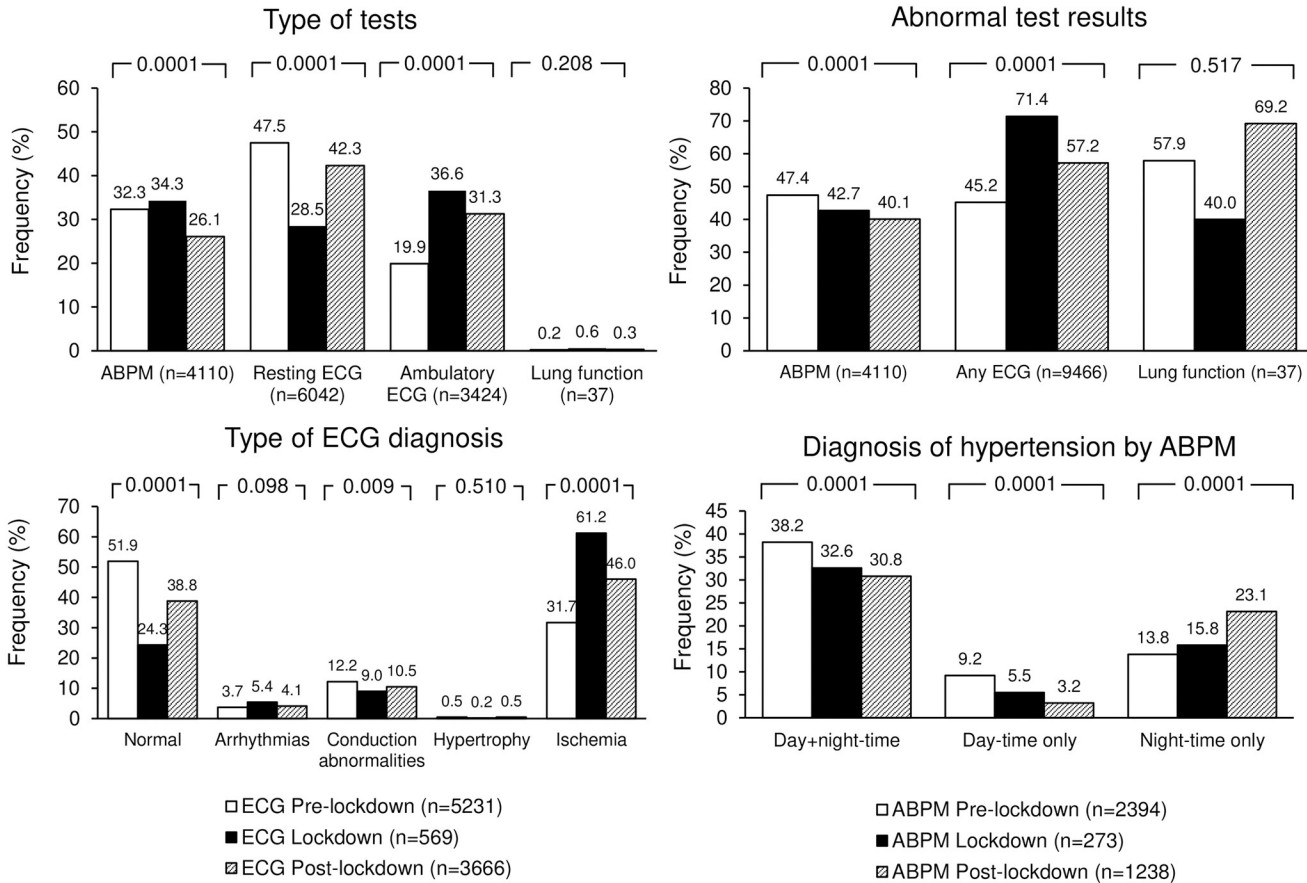

**Fig 3. Frequency of type of tests, abnormal tests, ECG diagnosis, and diagnosis of hypertension by ABPM before the lockdown (open bars), during the lockdown (full bars), and after the lockdown (striped bars).** P-values of the differences are reported on top of each group of bars. ABPM: Ambulatory Blood Pressure Monitoring; ECG: Electrocardiogram.

before the pandemic) and the adherence to treatment (77.0±21.9 vs. 67.9 ±39.1%) was observed, although the between-periods difference did not achieve statistical significance.

During the lockdown, a significant increase in the number of SpO$_2$ values and a significant reduction in the entries for body temperature, body weight, blood lipids, or glucose values was observed (Fig 4).

The proportion of abnormal BP and SpO$_2$ readings was significantly less during the lockdown. In contrast, a significant increase in the number of abnormal HR values was found (Fig 4). After the lockdown, the proportion of abnormal body weight or waist circumference values significantly increased compared to the lockdown and pre-lockdown periods. A similar trend, although non-statistically significant, was observed for blood lipids or glucose levels.

## Confirmatory analysis

As shown in S1 Table and S1 Fig, the number of tests performed in community pharmacies or general practitioners' offices during the 2020 outbreak was significantly less than during the same period of 2019. Patients visiting community facilities in the year preceding the pandemic were younger and had a lower cardiovascular risk level. The proportion of abnormal tests was significantly greater during the 2020 lockdown period than during the same period of 2019 (S1 Table and S2 Fig).

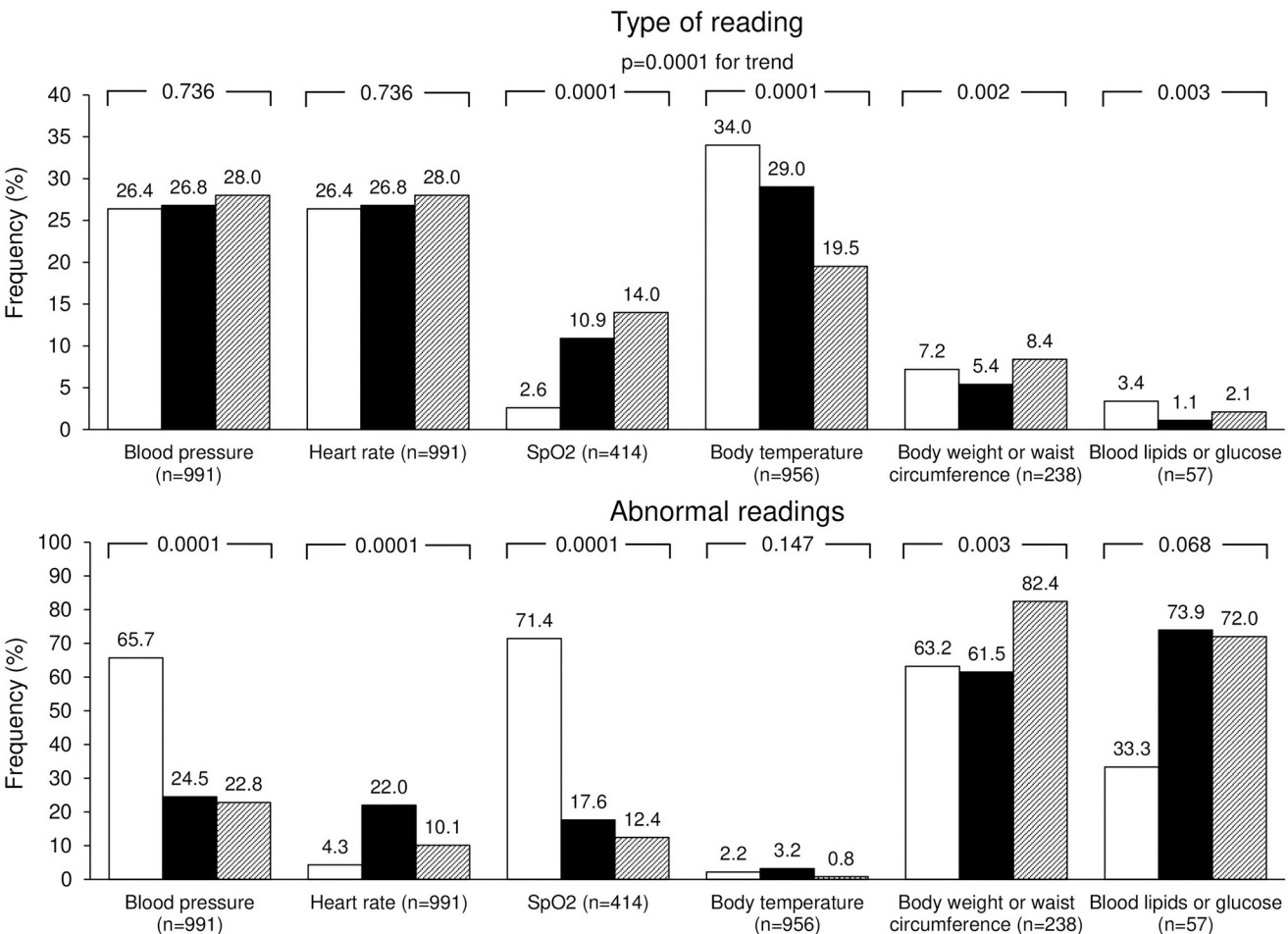

**Fig 4. Frequency of type of home readings and abnormal readings before the lockdown (open bars), during the lockdown (full bars), and after the lockdown (striped bars).** P-values of the differences are reported on top of each group of bars.

An analysis of the various types of tests showed an increment in the number of ECGs compared to ABPMs and lung function tests during 2020 than during 2019 (S3 Fig). The proportion of abnormal ECGs was much larger during the 2020 lockdown than during the corresponding period of 2019 (S2 Table and S3 Fig). BP control and results of lung function tests were better during the pandemic.

The number of home users and readings transmitted was significantly greater during 2020 than in 2019 (S1 Table). SpO2, body temperature, and body weight or waist circumference values were recorded more frequently during the pandemic than during the preceding year. In contrast, BP, HR, and blood lipids or glucose were less likely to be recorded during the pandemic than in the same period of 2019 (S4 Fig). In general, the proportion of abnormal values recorded at home was less during the pandemic (S2 and S5 Figs).

## Discussion

Our prospective observational study found a dramatic drop in the number of patients presenting to Italian community pharmacies or general practitioners' offices to carry out a professional diagnostic test during the containment period imposed by the 2020 recent coronavirus

outbreak. The patients were older and more frequently affected by cardiovascular disease than those presenting at the same facilities during the months preceding the pandemic. They also had a higher chance of reporting an impaired ECG, primarily due to a diagnosis of myocardial ischemia. In contrast, patients submitted to ABPMs showed a better 24-hour BP control and a higher frequency of antihypertensive drug treatment. However, the improved 24-hour BP control resulted from a balance between a better day-time BP and a worsened night-time BP control, particularly in the pandemic's post-lockdown phase. This finding was allegedly related to a combination of poor sleep quality during the hot summer and reduced frequency of treated individuals. Interestingly, following the withdrawal of containment measures, the number of patients visiting the pharmacies and general practitioners' offices rose. However, the patients' demographic and clinical characteristics did not substantially change, and the frequency and features of abnormal ECG tests remained very close to those of the lockdown period.

The increased prevalence of myocardial ischemia at the ECG observed in our study is neither surprising nor new. A possible explanation lies in that patients could not easily access the hospital for a proper diagnosis or follow-up or were reluctant to seek medical attention at the outpatient clinics or emergency rooms for fear of COVID-19. This is confirmed by evidence from the literature, showing a significant increase in the rate of out-of-hospital cardiac arrests and a decline in hospitalization rates for heart disease during the pandemic, but with patients showing more severe symptoms on admission and an excess of deaths [3, 34–39]. Our results also highlight telehealth's usefulness to timely identify and promptly refer to the hospital these patients in times of home isolation and social distancing [23, 40].

In contrast to what was observed in community pharmacies and general practitioners' offices, the number of home users and the amount of data exchanged between patients and doctors from home increased during the containment period. During this period, the prevalence of abnormal readings collected at home was lower than in the week preceding the isolation. During the lockdown, a significant increase in the number of $SpO_2$ values recorded was observed, likely because people in isolation sought a way to check their COVID-19 risk. Unexpectedly, body temperature, another parameter highly related to the COVID-19 risk, was less frequently collected at home during the outbreak. This discrepancy may be explained by the fact that subjects are usually less familiar with the interpretation of $SpO_2$ than with that of body temperature. Since the app returned immediate feedback of all inputted data's normality or abnormality level, users may have found this option particularly useful for $SpO_2$, but not for body temperature.

Interestingly, home BP control during and after the lockdown was better than before the pandemic. This contrasts with the evidence of a raised risk of high BP during the COVID-19 lockdowns available in a few reports [41, 42]. However, at variance from other studies, where patients were isolated and missed regular doctor appointments and contacts, our study patients were connected to their doctor. During the pandemic, their doctors checked their BP levels and promptly initiated an antihypertensive treatment or adjusted an ongoing treatment when it was deemed necessary. These patients confined at home also had improved adherence to their medication program, a well-known telehealth effect [43]. This finding was confirmed by the fact that ambulatory BP measured in community pharmacies or general practitioners' offices was also well controlled during the lockdown.

Another interesting finding of our study relates to the increased prevalence of overweight or obesity after the lockdown and abnormal blood lipids or glucose profiles during and after the lockdown. This might have been an unfavorable effect of the sedentary habit and unhealthy diet during the home isolation and a condition of anxiety and fear, which may also justify the increased prevalence of elevated HR values.

A confirmatory analysis including subjects managed at the same premises and home during the corresponding period in 2019 confirmed that the findings during the pandemic were not by chance.

## Strengths and limitations

Our study is one of the few analyzing telehealth's impact at scale in the community for managing patients with a chronic condition during the COVID-19 [44, 45]. Our findings highlight telehealth's importance in timely detecting abnormalities in patients' health status during a containment situation due to the pandemic and implementing the necessary interventions promptly.

Our research has some limitations that are mainly incident to its observational nature. First, all patients performed one single test in the pharmacy or general practice settings, and thus we could not track the outcome for any given patient before, during, and after the lockdown. The same is true for data collected at home. In this case, only one patient measured his BP before, during, and after the lockdown, and very few other patients had measures taken during and after the lockdown. Second, we do not know what happened to patients in the long term. However, we are currently performing a new analysis on data collected in the months following the first wave to verify this aspect. Third, we could have missed some relevant patients' clinical information as the users might not have provided all the required data when interviewed by the healthcare professional or through the app. Fourth, we must acknowledge that the sample of subjects measuring their parameters at home was limited in size, and thus our results may not be conclusive. Fifth, since home users manually inputted their data in the app and did not use wireless devices connected to the app, we cannot exclude that this approach may have limited the use of the app among aged subjects due to the likelihood of physical and/or intellectual limitation. Furthermore, we cannot rule out potential reporting bias during manual data entry. However, we carefully checked the occurrence of any potential outlier among the home data and corrected any incongruence to ensure data consistency.

Nevertheless, all these limitations are common to pragmatic studies that more closely reflect how patients are managed or are behaving in daily general practice.

## Perspectives

By and large, our results support the importance of providing surveillance through telehealth to patients with a chronic condition and the importance of community pharmacies and general practitioners' offices during a pandemic to ensure continuity of care of these patients and prompt referral to the necessary medical care. Such an approach may also be functional following the pandemic, particularly in countries like Italy with substantial regional medical care variation. Our study also suggests that it is urgent to implement a more significant number of telehealth solutions on the territory, promote better use of these services, and integrate them in the armamentarium of healthcare services provided to the population at risk, not only at the time of a pandemic. In order to accomplish this goal, future studies must be planned that will investigate the best model to be employed, particularly for home monitoring. Attention must be paid to provide user-friendly apps that can be affordable also for subjects at an advanced age with poor informatics literacy. The use of cheap wireless devices connected to smartphone apps may facilitate collecting and exchanging accurate data between patients and doctors and limit reporting bias. Finally, the app should not limit the monitoring to vital and non-vital parameters, but it should also be used to ensure treatment delivery and tracking of adherence to treatment protocols, functions particularly useful in times of pandemic and home isolation.

## Conclusions

Our study results at scale support the usefulness of a telehealth solution to monitor patients with a chronic condition and to detect deterioration of their health status during the COVID-19 pandemic.

## Supporting information

**S1 Fig. Weekly number of professional tests, home data, new visitors to the telemedicine website, and new app subscribers during the 2020 pandemic (thick continuous line) and during the corresponding period of 2019 (dashed line).** Pre-lockdown, lockdown and post-lockdown periods are identified as in Fig 1.
(TIF)

**S2 Fig. Weekly frequency (%) of abnormal tests in pharmacies and general practitioners' offices or abnormal readings collected at home during the 2020 pandemic (full bars) and during the corresponding period of 2019 (open bars).** Pre-lockdown, lockdown and post-lockdown periods are identified as in Fig 2.
(TIF)

**S3 Fig. Frequency of type of tests and abnormal tests during the 2020 pandemic (striped bars) and during the corresponding period of 2019 (open bars).** Pre-lockdown, lockdown and post-lockdown periods are identified. P-values of the differences are reported on top of each group of bars. ABPM: Ambulatory Blood Pressure Monitoring; ECG: Electrocardiogram.
(TIF)

**S4 Fig. Frequency of type of home readings during the 2020 pandemic (striped bars) and during the corresponding period of 2019 (open bars).** Pre-lockdown, lockdown and post-lockdown periods are identified. P-values of the differences are reported on top of each group of bars.
(TIF)

**S5 Fig. Frequency of abnormal home readings during the 2020 pandemic (striped bars) and during the corresponding period of 2019 (open bars).** Pre-lockdown, lockdown and post-lockdown periods are identified. P-values of the differences are reported on top of each group of bars.
(TIF)

**S1 Table. General characteristics of subjects and tests performed in the community before (2019) and during (2020) the COVID-19 pandemic.** Data are shown according to the setting (pharmacies or general practitioners' offices vs. home) and period (before, during, and after the lockdown).
(DOCX)

**S2 Table. General characteristics of subjects and tests performed in the community before (2019) and during (2020) the COVID-19 pandemic.**
(DOCX)

**S1 Checklist. TREND statement checklist.**
(PDF)

## Acknowledgments

The authors wish to thank all the pharmacists and general practitioners providing patients' data for the present publication.

## Author Contributions

**Conceptualization:** Stefano Omboni.

**Data curation:** Stefano Omboni.

**Formal analysis:** Stefano Omboni.

**Methodology:** Stefano Omboni.

**Writing – original draft:** Stefano Omboni.

**Writing – review & editing:** Tommaso Ballatore, Franco Rizzi, Fernanda Tomassini, Edoardo Panzeri, Luca Campolo.

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
