## [Decision Letter · Decision Letter 0]

10 Sep 2021

We look forward to receiving your revised manuscript.

Kind regards,

Luigi Lavorgna

Academic Editor

PLOS ONE

Journal Requirements:

Reviewers' comments:

Reviewer's Responses to Questions

**Comments to the Author**

1. Is the manuscript technically sound, and do the data support the conclusions?

Reviewer #1: Partly

Reviewer #2: Yes

2. Has the statistical analysis been performed appropriately and rigorously? 

Reviewer #1: N/A

Reviewer #2: Yes

3. Have the authors made all data underlying the findings in their manuscript fully available?

Reviewer #1: Yes

Reviewer #2: Yes

4. Is the manuscript presented in an intelligible fashion and written in standard English?

Reviewer #1: Yes

Reviewer #2: Yes

5. Review Comments to the Author

Reviewer #1: “Home data were collected with patients' own devices and were manually inputted in the mobile app.”

In my opinion, automatic data transmission would have eliminated reporting bias of manual entry. It would have been a very useful feature for the elderly due to the likelihood of physical and/or intellectual limitation.

It is not clear who inputted data in the mobile app, the patients? In this case, there could be a reporting bias of manual entry? Further, elderly patients could have more difficulties. In my opinion, this could have influenced the results. Therefore, this issue should be mentioned and discussed in the limits section of the manuscript.

In table 1: “Age (years)” correct as “Age (years, SD)”

In the results section: “During the lockdown, a significant increase in the number of SpO2 values and a significant reduction in the entries for body temperature, body weight, blood lipids, or glucose values was observed.” The authors discussed as: “During the lockdown, a significant increase in the number of SpO2 values recorded was observed, likely because people in isolation sought a way to check their COVID-19 risk.” Why did the authors observe a significant reduction in the entries for body temperature, considering that it is the first step to check the risk of COVID-19 infection? It should be discussed. In general, the authors should discuss more deeply all their results.

“In Italy, in the course of the first wave of the COVID-19 pandemic, available telehealth solutions have been used to track and promptly detect potentially infected individuals and monitor patients with any critical chronic condition restrained at home. However, preliminary reports suggest that the Country was mostly unprepared to implement existing telehealth solutions for this purpose.”

However, telemedicine for the management of chronic disease (behind cardiovascular diseases) was proposed and applied in several fields of medicine. For example, there are several chronic neurological diseases for which different telemedicine tools were proposed and tested (PMID: 34142263; PMID: 33802029; PMID: 33025327; PMID: 34173087; PMID: 34436726). I would add a sentence to mention this.

Reviewer #2: Interesting extensive observation. ables and figures well presented. Lack of follow-up is the major limitation. in Conclusions authors say: "the study suggests that it is urgent to implement a more significant number of telehealth solutions on the territory", could the authors give some suggestions or comment this point in Discussion? Considering recent implement in the approach with in-house therapy, could the author imagine a future project to support patients in this sense using their protocolo ?

6. PLOS authors have the option to publish the peer review history of their article (what does this mean?). If published, this will include your full peer review and any attached files.

Reviewer #1: No

Reviewer #2: No

---

## [Author Response · Author response to Decision Letter 0]

13 Sep 2021

Reviewer #1: “Home data were collected with patients' own devices and were manually inputted in the mobile app.” In my opinion, automatic data transmission would have eliminated reporting bias of manual entry. It would have been a very useful feature for the elderly due to the likelihood of physical and/or intellectual limitation. It is not clear who inputted data in the mobile app, the patients? In this case, there could be a reporting bias of manual entry? Further, elderly patients could have more difficulties. In my opinion, this could have influenced the results. Therefore, this issue should be mentioned and discussed in the limits section of the manuscript.

We thank the Reviewer for her/his positive evaluation of our work. We agree with the Reviewer. Indeed, the app can connect via Bluetooth to some devices that, however, need to be purchased. Unfortunately, no subject made use of these devices. Rather, they used their own devices. We added a sentence in the methods to clarify this point (page 6, lines 136-139).

The patient inputted the data, and, as correctly highlighted by the Reviewer, the risk of a reporting bias was potentially high. However, we made a check of potential outliers and corrected them. We have now discussed this aspect in the limitation section of the discussion (page 16, lines 350-355).

In table 1: “Age (years)” correct as “Age (years, SD)”

We thank the Reviewer for noticing this omission. As per her/his request, we have now amended Table 1 (but also Table S1 and S2) by adding “mean±SD” to age. 

In the results section: “During the lockdown, a significant increase in the number of SpO2 values and a significant reduction in the entries for body temperature, body weight, blood lipids, or glucose values was observed.” The authors discussed as: “During the lockdown, a significant increase in the number of SpO2 values recorded was observed, likely because people in isolation sought a way to check their COVID-19 risk.” Why did the authors observe a significant reduction in the entries for body temperature, considering that it is the first step to check the risk of COVID-19 infection? It should be discussed. In general, the authors should discuss more deeply all their results.

The Reviewer is correct. However, the finding is not surprising. The app returns immediate feedback on the normality of the value (see page 6, lines 124-125 “In the case of data collected at home, the mobile app provides immediate feedback to the user about the parameter's critical level”). Since subjects are familiar with the interpretation of body temperature but not with that of SpO2 the preferential use in the case of SpO2 may be linked to this app’s function, which was found particularly useful by the users. We acknowledge that this point was not adequately described and discussed. The discussion has been expanded in order to clarify this aspect (page 14, lines 309-314). Furthermore, the entire discussion has been revised and improved following both Reviewers’ suggestions.

“In Italy, in the course of the first wave of the COVID-19 pandemic, available telehealth solutions have been used to track and promptly detect potentially infected individuals and monitor patients with any critical chronic condition restrained at home. However, preliminary reports suggest that the Country was mostly unprepared to implement existing telehealth solutions for this purpose.”

However, telemedicine for the management of chronic disease (behind cardiovascular diseases) was proposed and applied in several fields of medicine. For example, there are several chronic neurological diseases for which different telemedicine tools were proposed and tested (PMID: 34142263; PMID: 33802029; PMID: 33025327; PMID: 34173087; PMID: 34436726). I would add a sentence to mention this.

The introduction has been revised and the original sentence has been modified in order to comply with Reviewer’s suggestion (page 3, lines 71-75). Wee have also added relevant references (from 14 to 22).

Reviewer #2: Interesting extensive observation. Tables and figures well presented. Lack of follow-up is the major limitation. in Conclusions authors say: "the study suggests that it is urgent to implement a more significant number of telehealth solutions on the territory", could the authors give some suggestions or comment this point in Discussion? Considering recent implement in the approach with in-house therapy, could the author imagine a future project to support patients in this sense using their protocol?

We thank the Reviewer for her/his positive evaluation of our work for her/his important suggestion. We have now added a section Perspectives and provided some consideration about the future of telehealth in this field. The section regarding conclusions has also been modified.

---

## [Editor Report · Decision Letter 1]

16 Sep 2021

Telehealth at scale can improve chronic disease management in the community during a pandemic: an experience at the time of COVID-19

PONE-D-21-21512R1

We’re pleased to inform you that your manuscript has been judged scientifically suitable for publication and will be formally accepted for publication once it meets all outstanding technical requirements.

Kind regards,

Luigi Lavorgna

Academic Editor

PLOS ONE
---

## [Editor Report · Acceptance letter]

17 Sep 2021

PONE-D-21-21512R1 

Telehealth at scale can improve chronic disease management in the community during a pandemic: an experience at the time of COVID-19 

Dear Dr. Omboni:

I'm pleased to inform you that your manuscript has been deemed suitable for publication in PLOS ONE. Congratulations! Your manuscript is now with our production department. 

Kind regards, 

on behalf of

Dr. Luigi Lavorgna 

Academic Editor

PLOS ONE